# Isolation of Antidiabetic Withanolides from *Withania coagulans* Dunal and Their In Vitro and In Silico Validation

**DOI:** 10.3390/biology9080197

**Published:** 2020-07-30

**Authors:** Saima Maher, M. Iqbal Choudhary, Farooq Saleem, Saima Rasheed, Imran Waheed, Sobia Ahsan Halim, Muhammad Azeem, Iskandar Bin Abdullah, Matheus Froeyen, Muhammad Usman Mirza, Sarfraz Ahmad

**Affiliations:** 1Department of Chemistry, Sardar Bahadur Khan Woman University, Quetta 95000, Pakistan; 2H.E.J. Research Institute of Chemistry, International Center for Chemical and Biological Sciences, University of Karachi, Karachi 75270, Pakistan; iqbal.choudhary@iccs.edu (M.I.C.); michealsorb@gmail.com (S.R.); 3Department of Chemistry, College of Science, King Saud University, Riyadh-11451, Saudi Arabia; 4Faculty of Pharmacy, The University of Lahore, Lahore 54000, Pakistan; farooq.saleem@pharm.uol.edu.pk (F.S.); muhammad.azeem@pharm.uol.edu.pk (M.A.); 5Akhtar Saeed College of Pharmaceutical Sciences, Bahria Town, Lahore 54000, Pakistan; imran.waheed@amdc.edu.pk; 6Natural and Medical Sciences Research Center, University of Nizwa, Nizwa 616, Oman; sobia_halim@unizwa.edu.om; 7Department of Chemistry, Faculty of Sciences, University Malaya, Kuala Lumpur 50603, Malaysia; iskandar.a@um.edu.my; 8Department of Pharmaceutical and Pharmacological Sciences, Rega Institute for Medical Research, Medicinal Chemistry, University of Leuven, B-3000 Leuven, Belgium; mathy.froeyen@kuleuven.be (M.F.); muhammadusman.mirza@kuleuven.be (M.U.M.)

**Keywords:** *Withania coagulans*, Solanaceae, withacogulanoside-B, antiglycation activity, α-glucosidase inhibition, molecular docking

## Abstract

*Withania coagulans* (*W. coagulans*) is well-known in herbal medicinal systems for its high biological potential. Different parts of the plant are used against insomnia, liver complications, asthma, and biliousness, as well as it is reported to be sedative, emetic, diuretic, antidiabetic antimicrobial, anti-inflammatory, antitumor, hepatoprotective, antihyperglycemic, cardiovascular, immuno-suppressive and central nervous system depressant. Withanolides present in *W. coagulans* have attracted an immense interest in the scientific field due to their diverse therapeutic applications. The current study deals with chemical and biological evaluation of chloroform, and *n*-butanol fractions of *W. coagulans*. The activity-guided fractionation of both extracts via multiple chromatographic steps and structure elucidation of pure isolates using spectroscopies (NMR, mass spectrometry, FTIR and UV-Vis) led to the identification of a new withanolide glycoside, withacogulanoside-B (**1**) from *n*-butanol extract and five known withanolides from chloroform extract [withanolid J (**2**), coagulin E (**3**), withaperuvin C (**4**), 27-hydroxywithanolide I (**5**), and ajugin E (**6**)]. Among the tested compounds, compound **5** was the most potent *α*-glucosidase inhibitor with *IC_50_* = 66.7 ± 3.6 µM, followed by compound **4** (*IC_50_*: 407 ± 4.5 µM) and compound **2** (*IC_50_*: 683 ± 0.94 µM), while no antiglycation activity was observed with the six isolated compounds. Molecular docking was used to predict the binding potential and binding site interactions of these compounds as *α*-glucosidase inhibitors. Consequently, this study provides basis to discover specific antidiabetic compounds from *W. coagulans*.

## 1. Introduction

*Withania* is a small genus that belongs to a diverse family of shrubs Solanaceae (comprising approximately 2000–3000 species distributed across 90 genera) [1]. Different species of *Withania* are widely distributed across the East Mediterranean to South Asian regions. Pakistan is home to its two species *Withania coagulans* and *W. somnifera* [2].

The genus *Withania*, particularly *W. coagulans*, is well-known for its diverse biological potential. According to the literature, different parts of the plant are used against impotence, wasting diseases, failure to thrive in children, insomnia, and nervous exhaustion. For instance, the active principle of its seeds is used in traditional medicines, and is responsible for milk coagulation [3]. Similarly, the sweet fruit of this plant has been found to be helpful against liver complications, asthma, and biliousness, as well as, it is found to be emetic, diuretic, and sedative. The flowers of *W. coagulans* are used in the treatment of diabetes. Apart from ethnobotanical applications, several therapeutic applications of this plant have been reported, including antihyperglycemic, anti-inflammatory, antitumor, antimicrobial, hepatoprotective, cardiovascular, immuno-suppressive, free radical scavenging, and central nervous system depressant activities [4,5,6]. Moreover, antimutagenic, antidiabetic, and leishmanicidal activities have also been described for this plant [7,8].

The major bioactive phytoconstituents isolated from *W. coagulans* are lactone steroids called as withanolides [9,10]. Withanolides have C-28 ergostane steroidal nucleus with *γ*- or *δ*-lactone, and a side-chain connected to the C-17 carbon atom [10]. According to the literature, withanolides have only been slightly explored in plants bearing withanolide and related ergostane-type steroids [11,12,13].

*α*-Glucosidase (EC3.2.1.20) is a small intestinal-membrane bound enzyme which catalyzes the hydrolysis of an oligosaccharide to absorbable monosaccharide, i.e.,, glucose, and thus its inhibition can suppress the postprandial hyperglycemia [14]. Therefore, inhibition of *α*-glucosidase is a useful intervention to manage type II diabetes [15]. Moreover, *α*-glucosidase antagonists have also been used as anti-obesity drugs, inhibitors of tumor metastasis, insect’s antifeedants, antiviral, and fungistatic compounds, and immune modulators [16].

Glycation is a chemical process in which biomolecules (such as DNA, lipids, and proteins) are modified by the non-enzymatic linkage of reducing sugars (e.g., glucose), resulting in highly reactive advanced glycation end products (AGEs). This process has been linked to some undesired health effects. e.g., diabetic complications, and aging. Thus, efforts are being made to discover new and safe antiglycation compounds to inhibit the formation of AGEs (e.g., dicarbonyl species, free radicals, protein cross-links, etc.) [17].

Molecular docking studies have discovered several novel scaffolds against specific biological targets in several studies [18,19]. We conducted docking of the newly isolated compounds on *Saccharomyces cerevisiae* (*S. cerevisiae*) *α*-glucosidase enzyme. Because the three-dimensional (3D) structure of *S. cerevisiae α*-glucosidase is not determined yet, we applied homology modeling to predict the 3D-structure of this protein before docking. Homology modeling has been applied previously to determine the structure and function of several challenging targets which cannot be explored via X-ray crystallography and NMR techniques due to their hydrophobic nature. Several *α*-glucosidase inhibitors have been reported based on homology model of *α*-glucosidase [20,21,22,23,24].

Based on antidiabetic and antiglycation potential of *W. coagulans* [25], current study was aimed to identify active constituents responsible for these activities. Additionally, the binding orientation of investigated compounds with α-glucosidase were investigated through molecular dynamics simulations and binding site residues responsible for inhibitor binding were also elucidated.

## 2. Experimental

### 2.1. General Experimental Method

Thin-layer chromatography (TLC) was carried out on pre-coated silica gel plates (Merck, Germany) and the spots were visualized initially under UV light at 254 nm and later sprayed with cerium (IV) sulfate reagent to develop the colors to predict different classes of natural products. The recycling high-performance liquid chromatography (HPLC) was employed for final isolation of pure compounds (JAI, LC-908W, Japan Analytical Industry Co. Ltd., Tokyo, Japan) with L-80 or ODS H-80 columns (YMC, Japan). UV spectra were recorded using a Shimadzu UV240 spectrophotometer in CH_3_OH for λ_max_ nm (log ε). The FTIR spectra were recorded using KBr disc method on a JASCO A-302 spectrometer and represented in cm^−1^. Proton (^1^H, 300 MHz) and carbon (^13^C, 100 MHz) NMR spectral measurements were performed in CD_3_OD on a Bruker AV-500 equipment with reference to the residual solvent signals (CD_3_OD/CDCl_3_), and the data was presented in δ (ppm). 2D NMR measurements were performed on a Bruker AMX 500 NMR spectrometer. EI-MS were performed at 70 eV on a Finnigan MAT-112 or MAT-312 equipment and main ions are represented as *m*/*z* (%). FAB-MS were measured in the glycerol matrix by using a JEOL HX-110 mass spectrometer.

### 2.2. Plant Material

Plant material was collected from Hub, Sindh, Pakistan and analyzed by Dr. Sher Wali Khan, Taxonomist, Department of Botany, University of Karachi. A voucher specimen was submitted in the same section (Voucher specimen No. 46257).

### 2.3. Extraction and Isolation

Air-dried whole plant of *W. coagulans* (3.4 kg) was macerated in methanol for 24 h, filtered and the solvent evaporated on a rotary evaporator under high vacuum. The extract obtained (*ca*. 120 g) was suspended in water and extracted with hexane to remove nonpolar constituents. The aqueous phase was extracted with chloroform at three pH levels (i.e. 3, 7 and 10) in ascending order [26]. The residual aqueous phase was extracted with butanol and water. The chloroform fraction extracted at pH 3 (19 g) was subjected to medium pressure liquid chromatography (MPLC) using chloroform/hexane gradient elution in the order of increasing polarity with 30, 70, and 100% chloroform resulting in three fractions i.e., MC1, MC2 and MC3, respectively. The fraction MC2 (6.3 g) was then subjected to column chromatography using silica gel and chloroform/hexane solvent system to elute several fractions and the pooling of similar fractions yielded seven fractions (A-G). The fraction A (2.3 g) was exposed to column chromatography with the same solvent system, followed by various chromatographic steps, such as Sephadex LH-20, polyamide and ODS column chromatography and further purified by RP-HPLC. This led to the isolation of compounds **2–6**.

The fractionation of *n*-butanol extract (31.5 g) was carried out on HP-20 column and the methanol/water (1:1) soluble portion was subjected to polyamide column chromatography using chloroform/methanol solvent gradient resulting in four subfractions [6]. The fraction WC2 obtained at 20% CHCl_3_: MeOH (3.8 gm) was subjected to silica gel column chromatography to afford five main fractions (A-E). Fraction C (0.3 g) was fractionated using silica gel column chromatography, with isocratic elution of 8% CHCl_3_:MeOH, and purification of RP-HPLC with L-80 column by using MeOH: H_2_O (1:1) at a flow rate of 3.5 mL/min yielded compound **1** (8 mg), and was characterized as a new compound.

3*β-*Hydroxy-14*R*,20*R*-epoxy-1-oxowitha-5,24-dienolide-27-*O*-*β-*D glucopyranoside (**1**)

White powder; C_34_H_48_O_11__,_ [α] ^25^_D_: -62 (*c_=_* 0.2, MeOH); UV (MeOH) λ_max_ nm 223 (unsaturated lactone); IR (KBr) *ν*_max_ cm^−1^: 1718 (carbonyl), 1684 (α,β-unsaturated lactone), 3490 (hydroxyl). ^1^H- and ^13^C-NMR spectroscopic data: (Table 1); HRFAB-MS *m*/*z* (+ve) *m*/*z:* [M + H]^+^ 633.3240 (calcd. for C_34_H_48_O_11_ = 633.3275).

### 2.4. Acidic Hydrolysis of Compound 1

Compound **1** (2 mg) was solubilized in 5% H_2_SO_4_/H_2_O (2 mL) and refluxed for 12 h. The reaction mixture was cooled to room temperature and extracted with ethyl acetate (1 × 3 mL) to get aglycons. The water layer was neutralized with aqueous sodium hydroxide and the presence of glucose was validated by co-TLC with authentic samples by using reported standard method [27].

### 2.5. In Vitro α-Glucosidase Inhibition Assay

*α*-Glucosidase inhibitory potential of test compounds was assayed by using 0.1 M phosphate buffer (pH 6.8) at 37 °C. The enzyme (0.2 U/mL) in phosphate buffer was incubated with selected concentrations of test compounds at 37 °C for 15 min. The substrate, *p*-nitrophenyl-*α*-d-glucopyranoside (PNP-G) (0.7 mM) was added in each well of 96-well plate and change in absorbance was monitored at 400 nm for 30 min. DMSO (7.5% final) and 1-deoxynojirimycin were used as negative and positive controls, respectively [28]. The increase in absorption due to the hydrolysis of PNP-G by α-glucosidase was recorded continuously with the spectrophotometer (Spectra Max, MD, USA.) and percent inhibition was calculated by using the following formula:$$\%\;\mathrm{Inhibition}\;=\;100-⦋\frac{\mathrm{OD}\;\mathrm{of}\;\mathrm{test}\;\mathrm{sample}}{\mathrm{OD}\;\mathrm{of}\;\mathrm{the}\;\mathrm{control}}\times 100⦌$$

### 2.6. In Vitro Antiglycation Assay

Methylglyoxal (14 mM), sodium azide (30 mM) in 0.1 M phosphate buffer, bovine serum albumin (10 mg/mL), and various concentrations of the compounds (prepared in 10% DMSO) were incubated under sterile conditions at 37 °C for 9 days. The samples were observed for the development of fluorescence (excitation, 330 nm; emission, 440 nm) against blank [28]. Standard rutin was used as a positive control and percent inhibition of AGE formation in the test sample *versus* control was calculated by using the following relation:$$\%\;\mathrm{Inhibition}\;=\;⦋1-\frac{\mathrm{Fluorescence}\;\mathrm{of}\;\mathrm{test}\;\mathrm{sample}}{\mathrm{Fluorescence}\;\mathrm{of}\;\mathrm{the}\;\mathrm{control}}⦌\times 100$$

### 2.7. Software Used/Statistical Analysis

The experiments were performed by using 96-well microplate reader (SpectraMax M2, Molecular Devices, CA, USA) and the results were analyzed by SoftMaxPro 4.8 and MS-Excel. The experiments were repeated in triplicate, and results were presented as means ± SEM. *IC*_50_ values were determined by using EZ-FIT, Enzyme kinetics software by Perrella Scientific, Inc., Amherst, MA, USA.

### 2.8. Homology Modeling and Molecular Docking

Homology modeling was conducted on the SwissModel server by using an automatic modeling mode. The sequence of *S. cerevisiae α*-glucosidase enzyme was downloaded from UniprotKB with the accession number P53341 for modeling. SwissModel server searched 3A47 and 3AXH as the best template structures. We used both templates for modeling *S. cerevisiae α*-glucosidase. Five models were obtained that were evaluated based on their stereochemical and geometrical properties by different servers, including Procheck, ERRAT and verify3D. The best model was selected for further study.

### 2.9. Molecular Dynamics Simulation and Binding Free Energy Calculations

Molecular docking was performed using AutoDock Vina [29] and best pose was subjected to 20 ns MD simulations. The MD simulations were carried out using AMBER simulation package 18 [30]. The same MD simulation protocol was implemented as described in previous studies [31,32,33]. The topology and coordinate files of complex was generated through tleap program of AMBER, and Antechamber package of AmberTools was utilized and parameters were taken from the GAFF force field (GAFF) [34], After stepwise minimization, the simulation system was heated and equilibrated accordingly and a final production run of 20 ns was performed at standard temperature (300K) and pressure (1 bar). The trajectory analysis was performed using CPPTRAJ program of AMBER [35].

The MM-GBSA approach is well illustrated in binding free energy calculations [36,37,38,39] which was utilized to estimate the total binding free energy ΔG_tol_ of the complex which is the sum of the molecular mechanics binding energy (ΔE_MM_) and solvation free energy (ΔG_sol_) as given below:ΔE_MM_ = ΔE_int_ + ΔE_ele_ + ΔE_vdw_
ΔG_sol_ = ΔG_p_ + ΔG_np_
ΔG_tol_ = ΔE_MM_ + ΔG_sol_

## 3. Results and Discussion

### 3.1. Characterization of the Isolated Compounds

Compound **1** was isolated in the form of white amorphous powder. During structural elucidation its IR spectrum depicted absorptions at 3385, 1735, 1660, and 1593 cm^−1^, demonstrating the presence of hydroxyl group, carbonyl ester, and C = C, respectively. The UV-Vis spectrum showed *λ*_max_ at 223 nm characteristic of conjugated enone and conjugated *δ*-lactone chromophores which usually exist in withanolides. The molecular formula of compound **1** was C_34_H_48_O_11,_ determined with the help of HR-FABMS (^+ve^) showed [M + H]^+^ at 633.3240 (Calc. for C_34_H_48_O_11_ +H = 633.3275).

The ^1^H-NMR spectrum of compound **1** showed four signals for 3H singlets at δ 1.35, 1.27, 1.59, and 2.01 which were assigned to the C-18, C-19, C-21, and C-28 methyl groups, respectively. The presence of AB doublets at δ 4.30, 4.31 (*J*_27α,27β_ = 11.4 Hz) implied that the C-27 position is substituted with an oxygenated group. The downfield chemical shift at δ 3.84 br, m, was assigned to C-3, its *W*_1/2_ (14.5 Hz) indicated that hydrogen is axially oriented. The characteristic double doublet at δ 4.23 (1H, dd, *J* = 13.2, 3.4 Hz) was assigned to the C-22 methine proton of the lactone moiety. The multiplicity of the H-22 signal indicated the absence of any proton at the vicinal C-20. A one-proton broad doublet at δ 4.58 (1H, br s) was assigned to the C-6 vinylic proton, while doublet at δ 4.36 (*J* = 7.6 Hz) was assigned to the anomeric proton of the sugar moiety.

^13^C-NMR spectra of **1** indicated that there were four methyl, ten methylene, eleven methine, and nine quaternary carbons, respectively. ^1^H and ^13^C-NMR spectrum (Table 1) indicated that there is an aglycone part of the compound similar to coagulin J [6] and a sugar moiety. The position of the sugar residue was resolved by using the HMBC correlation technique, in which C-27 (4.30, 4.38) showed correlations with anomeric carbon C-1′ (103.0) Key HMBC correlations in compound **1** is shown in Figure 1B. The stereochemistry at C-17 was inferred to be *R* with α-oriented ether bridge that occurred between C-14/C-20 [6]. The stereochemistry at other asymmetric centers of **1** was assigned by comparison with coagulin J [6] and NOESY spectrum (Figure 1C). Glucose was identified by hydrolysis followed by co-TLC with an authentic sample. The above analysis depicted the structure of **1** as 3*β*-hydroxy-14, 20-epoxy-1-oxowitha-5, 24-dienolide-27-*O*-*β-*d-glucopyranoside. The known compounds isolated from *W. coagulans*, were identified as withanolid J (**2**) [40], coagulin E (**3**) [6], withaperuvin C (**4**) [41], 27-hydroxywithanolide I (**5**) [42], and ajugin E (**6**) [43] by comparing spectral data to the reported data in the literature. The structures of the compounds **2**–**6** are presented in Figure 2.

### 3.2. Biological Activity

*α*-Glucosidase inhibitory activity of fractions from *W. coagulans* and its compounds were evaluated. Among two fractions, i.e., CHCl_3_ (crude) and BuOH (crude), only CHCl_3_ was found to be active with an *IC_50_* value of 0.2 ± 0.01 mg/mL. Among the tested hits, compound **5** was the most potent compound against the *α*-glucosidase enzyme with an *IC_50_* value of 66.7 ± 3.6 µM. The compound **4** and **2** depicted moderate to lower level of activities with *IC_50_* values of 407 ± 4.5, and 683 ± 0.94 µM, respectively, when compared with 1-deoxynojirimycin (i.e., standard inhibitor; *IC_50_* = 440.99 ± 0.01 µM). It was observed that compounds **1**, **3** and **6** did not possess *α*-glucosidase inhibitory activity. The activities of compounds are tabulated in Table 2. Fractions from *W. coagulans* and pure secondary metabolites were also screened against antiglycation activity. These fractions and compounds showed less than 50% inhibition; therefore, these compounds were considered as inactive.

Based on exhaustive literature search for potential biological activities of compounds closely related to isolates **1**–**6**, 17β-hydroxywithanolide K and 4β-hydroxywithanolide E were found potent antihyperglycemic and anti-inflammatory in Sprague-Dawley normoglycemic rat model [44].

### 3.3. Molecular Modeling and Simulations

Molecular docking was carried out to predict the structural requirements of compounds to inhibit the structure and function of *α*-glucosidase enzyme. For this purpose, we conducted homology modeling of *α*-glucosidase from *S. cerevisiae*. Initially, the template structure was searched on NCBI protein BLAST to model the protein of interest, and we obtained a crystal structure of isomaltase enzyme from *S. cerevisiae* (PDB code: 3A47) with 72% identity and 99% query coverage. The maximum score was 909 with E-value 0. Similarly, another structure of isomaltase enzyme in complex with isomaltose (PDB code: 3AXH) was retrieved from *S. cerevisiae* with 72% identity, 99% query coverage, 907 scores and E-value 0. Both structures were used to model *S. cerevisiae α*-glucosidase enzyme. The active site was predicted by superimposing the developed model on 3AXH. The stereochemical aspects of the model were inspected by procheck Ramachandran plot, verify-3D, and ERRAT plots. The Ramachandran plot statistics showed that the protein contains a total of 579 residues, among them 444 (86.7%) residues lie in the most preferred region, while 63 (12.3%) residues were situated in the additional allowed region. The number of residues present in the generously allowed region is 3 (0.6%), however, only two residues (0.4%) are present in disallowed regions. These residues are Ala278 and Thr566, both the residues are not a part of the active site, thus our model is of good quality. The homology model was further refined and optimized by small MD simulations of 20 ns to remove the clashes and to adjust overall geometry of the backbone. We obtained 93.52% quality factors from ERRAT, while 95.5% of residues presented an average 3D-1D score of 0.2 from verify3D, this suggests that the quality of the model is satisfactory and can be used in docking experiments. Homology modeling results are presented in Figure 3.

The overall structure of *α*-glucosidase was found similar to *α*-amlylases [45,46] and isomaltase [47]. The catalytic residues are conserved among *S. cerevisiae* Isomaltase and *α*-glucosidase. Glu276 and Asp214 of *S. cerevisiae α*-glucosidase act as proton donor and nucleophile, respectively. While Asp349 act as a transition state stabilizer for substrate molecule. The catalytic residues and the surrounding residues involved in the formation of the active site are tabulated in Table 3.

The generated model was used in docking studies. Molecular docking was carried out by AutoDock Vina. In this study, we only investigated the predictive binding potential of the most active compound **5** which indicated reasonable docking binding energy of −6.68 kcal/mol. As water molecules have prime importance in catalytic activity of this enzyme, therefore MD simulation was performed in explicit solvent environment. Later, the MMGBSA binding free energies were estimated in the presence of solvent effect. The results of MD simulations are interactively displayed in Figure 4. During MD simulation, compound **5** remained stable inside the binding pocket in close proximity of catalytic triad residues as depicted from a final snapshot obtained after 20 ns time scale (Figure 4A,B). Moreover, the stability of **5** was evident from the energy decomposition analysis of residues involved in the formation of the active site followed by H-bond occupancy over a period of 20 ns (Figure 4C,D). The MD simulated binding mode of **5** established four well-populated H-bonds with the sidechains of His239 (average, 1.7 Å; 75.5%) and Glu276 (average, 2.5 Å; 17.6%) and backbone atoms of Phe157 (average 2.08 Å; 62.3%), Arg312 (average 2.53 Å; 45.6%). Other than these, Asn237 and Asn241 also showed favorable H-bond occupancy of 40.9 and 23.8%. These molecular interactions were in-line with the reported experimental studies of f α-glucosidases [20,21,23,48,49,50].

To analyze the residual contribution in overall binding energy of complex, per-residue decomposition analysis was performed which indicated His239 and Arg312 as highly contributing residues in overall binding free energy with a ΔG_total_ of −3.12 and −2.89 kcal/mol that mainly interacted electrostatically together with Asn241 (ΔG_total_ = −1.39 kcal/mol) and Phe300 (ΔG_total_ = −1.42 kcal/mol) which mainly contributed through vdW interactions. All these residues were found highly contributing to stabilize the complex formation during MD simulation. While, the catalytic triad weakly contributed as indicated from ΔG_total_ values of Asp214 (−0.15 kcal/mol), Glu276 (−0.89 kcal/mol) and Asp349 (−0.45 kcal/mol). This was due to the contribution of unfavorable solvation energy by blocking out a sufficient amount of solvent that might stabilize these negatively charged catalytic triad residues. The root-mean-square-deviation (RMSD) and root-mean-square-fluctuations (RMSF) was also examined over a period of 20 ns which revealed consistent backbone stability including the active site residues (Figure 4E) which contributed well in stabilizing the complex (Figure 4F,G).

## 4. Conclusions

In the present study, a phytochemical investigation was carried out on CHCl_3_ and *n*-butanol fraction of *W. coagulans* plant. All the compounds **2–6** isolated from chloroform fraction were known, while a new compound (i.e., compound **1**) was isolated from *n*-butanol fraction. It was observed that *n*-butanol fraction, and its isolated new compound were inactive in both antiglycation, and *α*-glucosidase inhibition assay, while CHCl_3_ fraction and its isolated pure compounds showed activity against *α*-glucosidase and were inactive in antiglycation assay. The activity was correlated with molecular docking studies that depict that the inactive compounds do not bind appropriately with the *α*-glucosidase enzyme. This study provides scientific basis to discover specific anti-diabetic compounds from *W. coagulans*.

## Figures and Tables

**Figure 1 biology-09-00197-f001:**
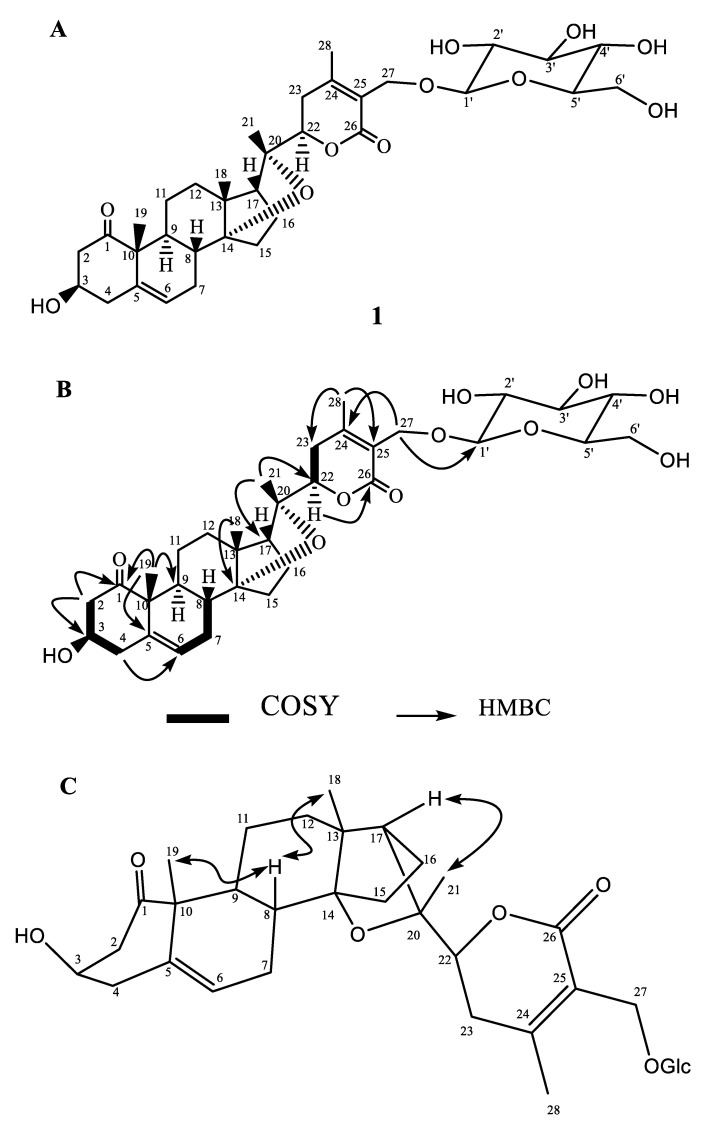
(**A**) Structure of compound **1**, (**B**) key COSY and HMBC interactions in compound **1**, (**C**) key NOESY interactions in compound **1**.

**Figure 2 biology-09-00197-f002:**
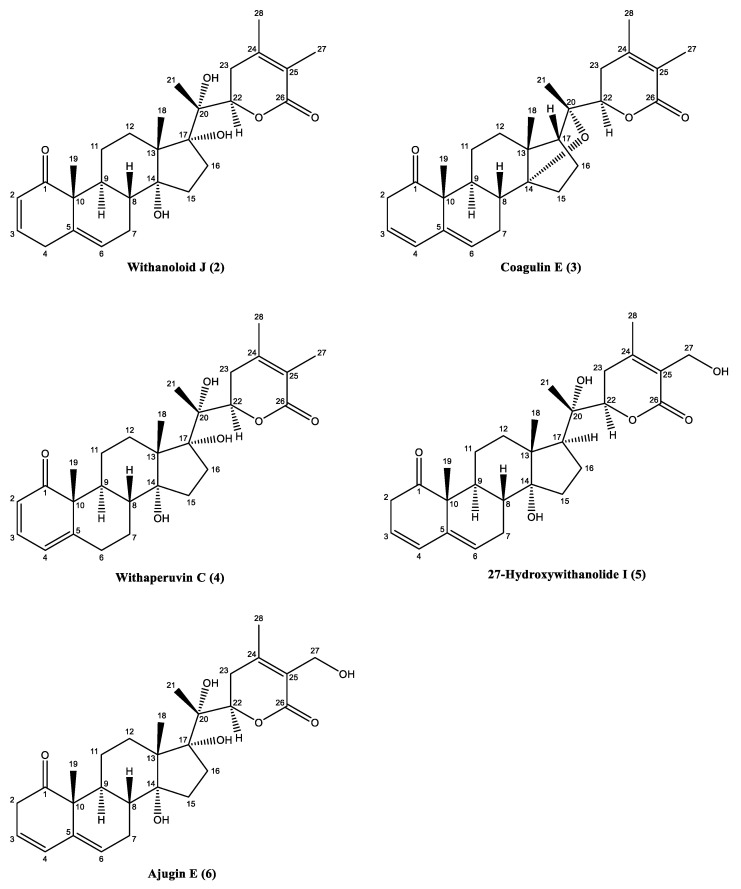
Structures of compounds **2** to **6** isolated from *w. coagulans*.

**Figure 3 biology-09-00197-f003:**
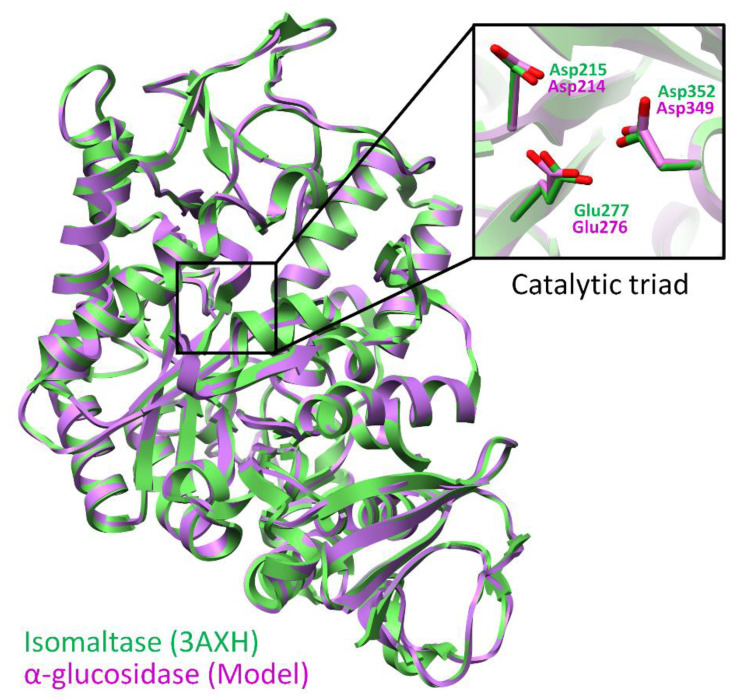
The superimposed view of model (purple ribbons) and template (3AXH in green ribbons) shows the structural topology of the model is similar to its template and residues of catalytic triad is represented in stick representation.

**Figure 4 biology-09-00197-f004:**
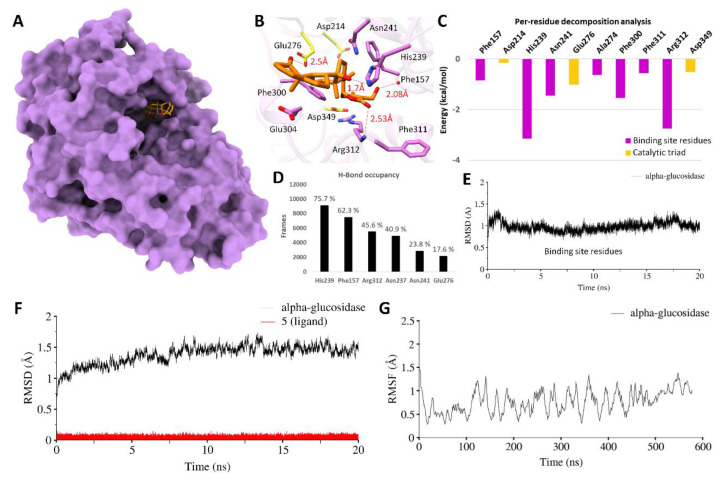
Molecular surface representation of *α*-glucosidase. (**A**) MD simulated binding pose of 5 (orange sticks) inside the binding pocket. Residues are highlighted in stick representation and catalytic triad residues are colored yellow. (**B**) H-Bonds with distance are displayed beside each bond. (**C**) Per-residue decomposition analysis of binding pocket residues, while catalytic triad residues are colored yellow. (**D**) H-Bond occupancy and (**E**) root-mean-square-deviation (RMSD) of binding pocket residues over a period of 20 ns. (**F**) RMSD of backbone atoms of *α*-glucosidase (in black) along with ligand (in red) and root mean square fluctuations and (RMSF) (**G**) are displayed over a period of 20 ns.

**Table 1 biology-09-00197-t001:** ^1^H- and ^13^C—NMR chemical shift values of compound **1** (ppm, CD_3_OD, 600 and 150 MHz respectively).

C. No.	δ_C_	δ_H_ (*J*, Hz)	C. No.	δ_C_	δ_H_ (*J*, Hz)
1	220.4	-	18	20.7	1.35 s
2	40.5	2.31, 2.12	19	18.2	1.27 s
3	75.0	3.84 br m, (W_1/2_ = 14.5)	20	78.1	-
4	41.0	2.04, 2.01 m	21	21.1	1.59
5	136.3	-	22	83.1	4.23 dd (13.2, 3.4)
6	123.6	4.58 br s	23	32.9	1.60, 1.84 overlap
7	32.6	1.77, 1.75 m	24	160.1	-
8	22.1	1.63 m	25	126.4	-
9	21.0	1.61 m	26	168.0	-
10	52.7	-	27	62.7	4.30 d (11.4), 4.31 d (11.7)
11	21.1	1.27, 1.35 m	28	18.0	2.01 s
12	20.8	1.61, 2.01 m	1′	104.0	4.36 d (7.6)
13	49.0	-	2′	75.0	3.31 t (8.1)
14	85.5	-	3′	74.0	3.65 overlap
15	22.3	1.63, 1.75 m	4′	71.6	3.33 overlap
16	32.8	1.35, 1.09 m	5′	78.0	3.33 overlap
17	50.5	2.30 t (9.4)	6′	63.5	3.64 dd (11.8, 4.8), 3.84 d (11.8)

**Table 2 biology-09-00197-t002:** α-Glucosidase inhibition and antiglycation activities of compounds **1**–**6**.

Compound	α-Glucosidase Inhibition	Antiglycation
	*IC_50_* (µM) ± (SEM)	*IC*_50_ (µM) ± (SEM)
**1**	Inactive	Inactive
**2**	683 ± 0.94	Inactive
**3**	Inactive	Inactive
**4**	407 ± 4.5	Inactive
**5**	66.7 ± 3.6	Inactive
**6**	Inactive	Inactive
Standard	440.99 ± 0.01 ^a^	288.9 ± 1.84 ^b^

^a^ Deoxynojirimycin, ^b^ Rutin.

**Table 3 biology-09-00197-t003:** Comparison of active site residues of template and model structures.

Template (*S. cerevisiae* Isomaltase)	Model (*S. cerevisiae α*-Glucosidase)
**Catalytic residues**	
Asp215	Asp214
Glu277	Glu276
Asp352	Asp349
**Extended active site residues**	
Asp69	Asp68
Tyr72	Tyr71
Val109	Val108
His112	His111
Tyr158	Phe157
Phe159	Phe158
Phe178	Phe177
Gln182	Gln181
Arg213	Arg212
Val216	Thr215
Gln279	Ala278
Phe303	Phe300
Arg315	Arg312
His351	His348
Gln353	Gln350
Glu411	Asp408
Arg442	Arg439
Arg446	Arg443
**Interacting water molecules**	
1021, 1026, 1056, 1058, 1061, 1087, 1102, 1122, 1174, 1228	1021, 1026, 1056, 1058, 1061, 1087, 1102, 1122, 1174, 1228

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
