# Peer review of "Isolation of Antidiabetic Withanolides from Withania coagulans Dunal and Their In Vitro and In Silico Validation"

_biology, 2020, doi:10.3390/biology9080197_

Round 1

Reviewer 1 Report

In the current study, Maher et al focused on the chemical and biological evaluation of chloroform and n-butanol fractions of Withania coagulans. In addition, these fractions of W. coagulans were tested for anti-α-glucosidase and antiglycation activity. The authors also isolated 6 compounds using spectroscopies (NMR, mass spectrometry, FTIR and UV-Vis). While the work has no major flaw in terms of experimental design, the language need to be extensively revised. In addition, the presentation style and consistency need to be enhanced to allow for easy flow of the content. Consider the following concerns based on my observation. Note that the annotated PDF also contain suggested changes for your attention.

Asbtract:Too much emphasis on the fractions, can be reduced

include a short sentence at the end to 'highlight' the implication of the findings.

Introduction: The flow need to be enhanced. Some of the paragraphs need to be revised. Ensure that the cohesion and logic of the different concepts are well-aligned and structured.

Aim of the study need to be explicitly presented as the last sentence in the introduction. Currently, not clear.

Result & discussion: Abit weak, authors Failed to extensively discuss the implication of their result. What was presented is a description of your results ONLY. See for e.g. section 3.2 which does not have a SINGLE citation. This is unacceptable and need to be revised.

Why so much emphasis on the docking aspect? This need to be justfiied.

References:Need more attention and generall editing. SOme of these have been highlighted below:

All the scientific names MUST be in italics

Consistency in the name of the journals. Full title vs abbreviation?

Appropriate Capitilization of first word for the name of each journal.

Reviewer 2 Report

The authors have isolated some of antidiabetic withanolides from Withania coagulans, involving a new compound that isolated from CHCl3 fraction, and analysis the in vitro bioactivity to determine the IC50 of inhibition of α-glucosidase and antiglycation. However, most of compounds were inactive and provided poor IC50 forα-glucosidase and antiglycation. In the other hand, authors should make sure identified data is completely consistence between Table 1 of text and experimental 3.1 section. For bioassay, author should provide appropriate standard inhibitor for Antiglycation in Table 2. Thus, I recommended the reject this paper, here are some corrections and suggestions detailed below:

  1. Author should make sure the calculation formula is simplified in section2.5.

Number of C of each structures and group should be significant marked in Figure 1.
